# Shared associations identify causal relationships between gene expression and immune cell phenotypes

Christiane Gasperi [1,5], Sung Chun [2,3,6], Shamil R. Sunyaev[2,3] & Chris Cotsapas [1,4✉]

Genetic mapping studies have identified thousands of associations between common variants and hundreds of human traits. Translating these associations into mechanisms is complicated by two factors: they fall into gene regulatory regions; and they are rarely mapped to one causal variant. One way around these limitations is to find groups of traits that share associations, using this genetic link to infer a biological connection. Here, we assess how many trait associations in the same locus are due to the same genetic variant, and thus shared; and if these shared associations are due to causal relationships between traits. We find that only a subset of traits share associations, with many due to causal relationships rather than pleiotropy. We therefore suggest that simply observing overlapping associations at a genetic locus is insufficient to infer causality; direct evidence of shared associations is required to support mechanistic hypotheses in genetic studies of complex traits.

[1] Department of Neurology, Yale School of Medicine, New Haven, CT, USA. [2] Division of Genetics, Brigham and Women's Hospital, Boston, MA, USA. [3] Department of Biomedical Informatics, Harvard Medical School, Boston, MA, USA. [4] Department of Genetics, Yale School of Medicine, New Haven, CT, USA. [5] Present address: Department of Neurology, Klinikum rechts der Isar, TUM School of Medicine, Technical University of Munich, Ismaninger Str. 22, Munich, Germany. [6] Present address: Division of Pulmonary Medicine, Boston Children's Hospital, Boston, MA, USA. ✉email: cotsapas@broadinstitute.org

Genetic mapping studies have identified thousands of associations between common variants and hundreds of human traits. Uncovering the mechanisms that underlie these traits requires understanding the molecular, cellular and physiological events altered by causal genetic variants. Incomplete fine mapping due to linkage disequilibrium and the possible action of causal variants across diverse cell types, contexts and genes currently limits our ability to infer the mode of action of causal variants, and hence the biology underlying traits. Experimentally testing multiple such mechanistic hypotheses across thousands of associations rapidly becomes a problem of scale; we thus need principled approaches to generating and testing such mechanistic hypotheses.

We and others have suggested such an approach, building on the concept of pleiotropy. The molecular and cellular events altered by causal genetic variants are, by definition, also genetic traits, and they must be associated with the same variant. To link traits together and thus form mechanistic hypotheses, one can thus look for shared genetic associations between traits[1–4]. Such sharing is often defined as two traits associated with variation in the same general genome region, often within an arbitrary window of physical distance. A more robust alternative is to identify pairs of traits that share an underlying causal effect, rather than a shared genomic segment, and several methods have been developed to this end[5–12].

We have reported a relative paucity of overlaps between expression quantitative trait loci (eQTLs) and disease risk associations[12]. This result appears paradoxical given the strong enrichment of risk heritability in gene regulatory regions[13–15], which suggests that the majority of risk effects should alter gene regulation, and therefore expression. This paucity may be because we are not interrogating the right cell types, or the right physiological or stimulation conditions for those cells; or it may be that we lack power to detect such overlaps, though our simulations suggest the latter is not a major factor[3,12].

It is tempting, therefore, to assume that requiring a demonstration of shared association between traits is overly stringent, and simply identifying associations to the same region is a more productive approach. A further temptation is then to assume causality, especially when the two traits are drawn from different levels of physiology: it is natural to assume that a gene expression trait is causal for disease risk rather than the other way around, for example. This assumption of causality is made implicitly when pairs of associations are used to propose mechanistic hypotheses of pathophysiology. However, it is also possible that the two traits are either associated to different variants in the same locus, or to a single pleiotropic variant and otherwise share no biological underpinning (horizontal pleiotropy). To our knowledge, there is no a priori way to set a prior expectation for causality or pleiotropy. How useful, therefore, is it to identify shared effects between two traits, rather than simply identify associations to the same broad locus?

Here, we answer this question by first assessing how many associations to different traits in the same locus are due to the same underlying effect, and thus shared; and if these associations shared between traits are likely due to a causal relationship between these traits, or if horizontal pleiotropy is widespread. We compare 164 distinct immune cell phenotypes from the Milieu Intérieur project[16,17] to gene expression traits in monocytes, neutrophils and T cells from the BLUEPRINT consortium[18]. We first select pairs of immune traits and gene expression traits with associations at the same genetic locus, and then identify which of these pairs share an association and which are associated with different variants in close proximity. We find that trait pairs with shared genetic associations are more likely to share a broader genetic correlation and are more likely to share a causal

relationship, as assessed by two Mendelian randomization approaches. Our results show that a substantial proportion of shared associations between traits is likely to be due to causal relationships. We therefore suggest that simply observing associations of different traits to the same genetic locus is insufficient to infer causality, and direct evidence of shared association must be the minimum evidence required to link traits.

## Results

**Immune phenotype loci harbor eQTLs but do not share associations with them.** To identify effects shared between immune and gene expression phenotypes, we first identified associations for each of the 164 immune phenotypes included in the Milieu Intérieur project data (Supplementary Table 1). JLIM, our joint likelihood mapping method, detects shared associations based on patterns of LD, so we excluded the major histocompatibility complex (MHC) locus on chromosome 6, where LD structure is particularly complex. We found 1379 distinct non-MHC loci (200 kilobases windows centered on the most associated variant) with evidence of independent association to at least one immune phenotype ($p < 1 \times 10^{-5}$; 32 loci at $p < 5 \times 10^{-8}$). In 83/1,379 loci, we found more than one independent effect for the same immune phenotype (conditional $p < 1 \times 10^{-5}$, 6% of loci).

We next looked for genes whose expression could plausibly be influenced by the same genetic effect in these loci. We found 14,634 genes with a transcription start site within 1 megabase of the lead variants, with at least one gene in 1318/1379 (95.6%) of the loci. We found that 7365/14,634 (50.3%) genes in 1201/1318 (91.1%) of loci have an eQTL in at least one immune cell type profiled by the BLUEPRINT Consortium ($p < 1 \times 10^{-3}$). Most of these 7365 genes were influenced by more than one conditionally independent eQTL within a 1.2 megabase window around the TSS: 3762/5748 (65.4%), 3476/5004 (69.5%), and 4146/5906 (70.2%) in T cells, neutrophils and monocytes, respectively (Supplementary Fig. 1). We included all these effects in our subsequent analyses, to make sure we capture all possible gene expression effects. Thus, consistent with previous reports across a spectrum of human traits[12,19], most loci associated with an immune phenotype also harbor at least one cis-eQTL, and many eQTLs have conditionally independent effects[20,21].

We used JLIM to assess if the immune phenotypes and nearby eQTLs were driven by the same underlying genetic effect, indicating shared mechanisms. After filtering, we compared 22,379 pairs of conditionally independent associations representing all 164 immune traits and 7060/7365 (95.6%) of genes, at 1199/1379 (86.9%) of the discovered loci. We found evidence for a shared underlying causal variant for 207/22,379 (0.9%) pairs, involving 92/164 (56.1%) immune phenotypes and 127/7060 (1.8%) distinct genes, at an FDR < 0.05. These 207 shared effects include 15 instances where a gene expression and immune phenotype pair shared two (13 pairs) or three (two pairs) genetic effects. For 190 distinct combinations of a gene expression trait and an immune phenotype at least one shared association could be identified. Thus, though we test the vast majority of cases where immune phenotypes and eQTLs overlap, we find limited statistical evidence for shared effects between them. This is consistent with our previous observations of limited sharing between autoimmune disease associations and cis-eQTLs[12]. Also similar to our previous findings, we observe that the power to detect shared association thus depends in part on statistical power in the secondary trait cohorts (Supplementary Fig. 2). We see correlation between immune traits across individuals, suggesting that some gene expression traits may spuriously share associations with more than one such trait; however, we found no difference in correlation between immune traits that shared

associations with gene expression traits and immune traits that did not (Spearman's $\rho = 0.11 \pm 0.13$ and $0.13 \pm 0.14$, respectively), suggesting this is not a major factor in our analysis.

Some of our results highlight clear relationships between traits: for example, we see a shared effect between expression level of *SELL* and the level of its protein product L-selectin (CD62L; Fig. 1a, b). We find the eQTL in all three BLUEPRINT cell types, and the immunological trait association in both neutrophils and eosinophils. As expected, the allele associated with increased *SELL* transcript abundance in neutrophils and monocytes is also associated with CD62L intensity in both neutrophils and eosinophils (Fig. 1a–c and Supplementary Fig. 3). However, the same allele decreases *SELL* transcript abundance in T cells. We find strong evidence that the T cell eQTL and CD62L intensity trait association are shared, suggesting a distinct mode of regulation in different cell types (for example, different factors binding to the same regulatory element in different cell types, with both types of interactions influenced by the same genetic variant).

We found that many shared effect *cis*-eQTLs are not in the same immune cell subpopulation as their cognate immune

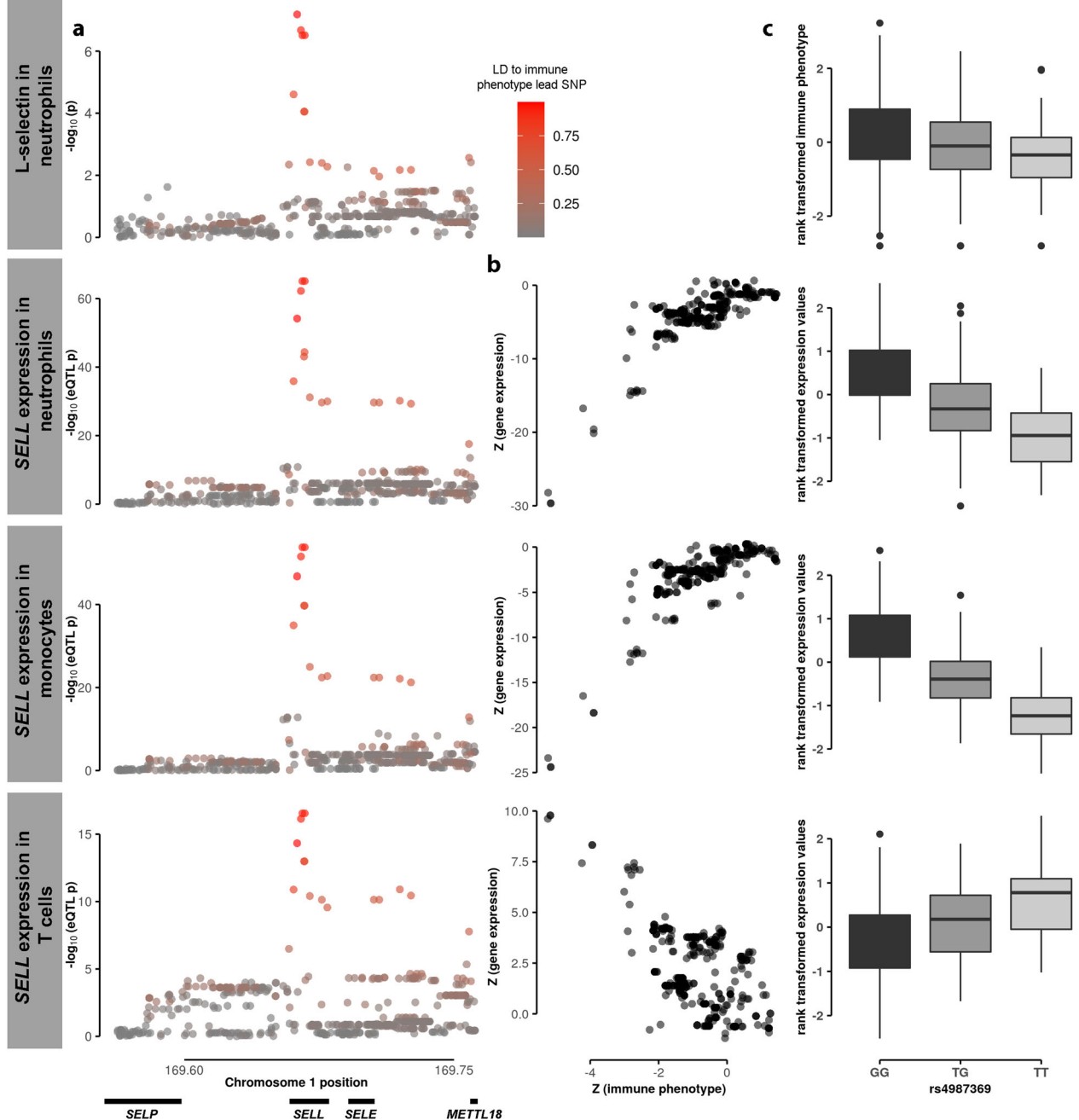

**Fig. 1 Shared genetic association on chromosome 1 for L-selectin (CD62L) in neutrophils and *SELL* expression in neutrophils, monocytes and T cells.** The association signal of the mean fluorescence intensity (MFI) of L-selectin (CD62L) in neutrophils at a genetic locus on chromosome 1 is consistent with the association signal to *SELL* expression in neutrophils, monocytes and T cells at the same genetic locus (**a**, shaded by linkage disequilibrium (LD, $r^2$) to the immune phenotype lead single nucleotide polymorphism (SNP)). The association Z statistics of the mean fluorescence intensity (MFI) of L-selectin in neutrophils and *SELL* expression in the three different cell types are strongly correlated (**b**). **c** The MFI of L-selectin and *SELL* expression in the three different cell types (both rank transformed) per genotype of the lead variant rs4987369.

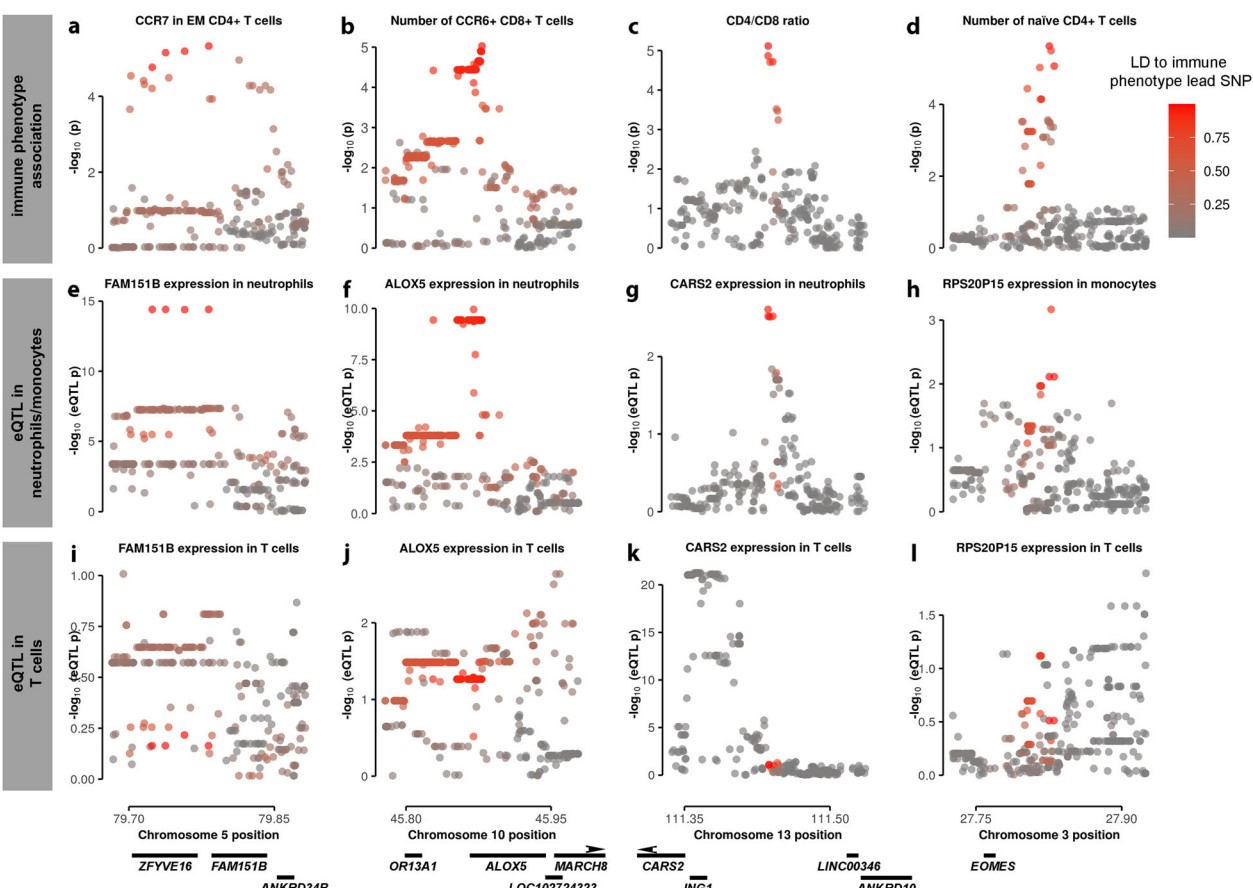

**Fig. 2 Shared genetic effects between gene expression traits in neutrophils or monocytes, but not in T cells, with immune phenotypes measured in T cells.** Shared association signals for different T cell immune phenotypes (**a–d**) and gene expression traits (expression traits quantitative loci, eQTLs) in neutrophils or monocytes (**e–h**). There were no consistent association signals for these gene expression traits in T cells (**i–l**). The genetic variants are shaded by linkage disequilibrium (LD, $r^2$) to the immune phenotype lead single nucleotide polymorphism (SNP).

phenotype. In some cases, we did not have expression data for the subpopulation in which an immunophenotype was measured. We found, for instance, a shared effect between the expression level of *CR2* in T cells and the level of its protein product, complement receptor type 2 or CD21 in multiple B cell populations (Supplementary Fig. 4). CD21 is the route through which Epstein-Barr virus infects B cells, and there is more recent evidence that this is also the mechanism of T cell infection[22]. This may therefore be a constitutive *CR2* eQTL, and may have a bearing on susceptibility to EBV infection.

We also found more complex patterns of sharing between cell types. We found several examples where a T cell immune parameter had a shared effect with an eQTL in monocytes or neutrophils, but the gene was either not expressed in T cells or there was no evidence of a shared eQTL in our data (Fig. 2). This suggests the possibility that changes to gene expression in one cell type have effects on population number and behavior of another cell type, consistent with the complex and dynamic interplay between immune cell subpopulations.

**Trait pairs sharing associations show broader genetic correlation.** Our broader goal is to establish whether shared associations can identify traits that are causally linked. We therefore sought to distinguish between horizontal pleiotropy, where the same variant influences two otherwise unrelated phenotypes; and mediation, where one phenotype is causal for the other. In horizontal pleiotropy, there should be no further genetic relationship between the traits. Conversely, in the case of mediation we expect

the two traits to share genetic architecture more broadly, as perturbation of the intermediate trait should have an effect on the outcome trait[23–25]. Therefore, to establish if traits with shared associations are more likely to be causally related, we first assess evidence for shared heritability between them, and then directly assess evidence for mediation.

To assess evidence for shared heritability between immune-expression phenotype pairs with a shared association, we asked if PRS for the gene expression trait in each pair predicts the immune trait. We compared 190 trait pairs with evidence of shared effects (92 immune and 127 expression traits in 116 loci) to 16,462 trait pairs that do not share the same causal variant per our JLIM analysis. For each trait pair, we calculated a genome-wide PRS for the gene expression trait, and determined the variance of the paired immune phenotype explained by that PRS (as $R^2_{PRS}$). We reasoned that if traits with a shared association are more likely to share heritability more broadly, we should see more variance explained in the 190 trait pairs than in the 16,462 that do not pass our JLIM analysis. We knew, however, that the presence of a shared association would bias this analysis in favor of our expected outcome, because we would be including a known positive association to the immune traits only in the 190 pairs. We accounted for this bias by conditioning the eQTL data on the main cis-eQTL effect with the strongest evidence for being shared with the immune phenotype, thus removing the effect of the shared variant (see Methods for details).

We found that the variance of an immune trait explained by gene expression PRS was higher when the two traits shared an

association than when they did not share one, even though we removed the effect of the shared association (Mann–Whitney–Wilcoxon $p < 0.05$; Table 1 and Fig. 3a, d). We found that this was generally true across a range of thresholds for selecting variants to include in the PRS calculation[26]. The proportion of trait pairs with a significant FDR adjusted empirical p-value in this analysis was higher for traits that shared an association (Fig. 3c, f, Table 1). We also found that in cases where an immune trait shared an association with more than one eQTL, including PRS for all shared eQTLs explained more variance than any single expression trait alone (Supplementary Fig. 5). Our analysis is conservative, as we condition on the variant with the strongest evidence of shared association; as expected, including lead cis-eQTL effects in the PRS calculations shows an even more extreme difference (Supplementary Table 2).

To ensure our results were not due to selection artefacts induced by p-value thresholds, we also examined the correlation between all JLIM p-values and variance of immune phenotypes explained by the gene expression PRS, and found significant correlation (Table 1, Fig. 3b, e). Together, the results of the PRS analyses provide evidence for a stronger genetic correlation between colocalized gene expression and immune phenotypes if they share the same underlying genetic effect.

**Trait pairs sharing associations are more likely to be causally related.** We next assessed evidence for causality directly with Mendelian randomization MR, again comparing trait pairs with a shared association to trait pairs with no evidence of shared associations. As our gene expression and immune phenotypes come from different cohorts, we used TSMR[27,28]. Statistical power in MR analyses is increased by selecting multiple variants as instruments[29]. This presents a problem when considering eQTLs as intermediate traits, because *cis*-acting effects often explain a large portion of phenotypic variance, and there is little power to detect *trans*-acting effects genome-wide[30–32]. To account for this power issue, we selected 16,631/16,652 trait pairs from above (190 with, and 16,441 without a shared association to an immune phenotype) where we found evidence for at least 2 independent variants associated with the gene expression trait (conditional $p < 1 \times 10^{-5}$). Using these variants as TSMR instruments, we found that the estimated causal effects of gene expression traits on immune phenotypes were higher in the 190 pairs with a shared association compared to the 16,441 non-sharing pairs (Fig. 4a), but this difference was not significant. However, we observed overall correlation between JLIM p-values and the magnitude of the causal effect of gene expression traits on immune phenotypes (Fig. 4b). We did not observe any trait pair with a p-value <0.05 after FDR correction in this analysis; however, the proportion of trait pairs with an unadjusted p-value <0.05 was higher for trait pairs with a shared association (Fig. 4c). These results suggest that trait pairs sharing an association are more likely to be causally related, and that this likelihood increases with increasing evidence that a shared association exists.

To address the limited number of instruments available for gene expression traits, we broadened our analysis to include other transcripts in the locus influenced by the same variants[33]. For each trait pair, we first identify variants independently associated with the gene expression trait, as above. We then ask if these variants are associated with any other transcript levels in the locus (within 1 megabase of the lead variant in the shared association test), and if so, identify all variants independently associated with those transcripts too. We thus gather a larger set of instruments for our MR analysis. We remove transcripts whose overall expression is highly correlated to the initial gene expression trait, to avoid over-estimating the effects of variants, as previously

**Table 1 Expression quantitative trait locus (eQTL) variants explain more immune trait variance when they share an association.**

| eQTL significance threshold (p-value) | Number of variants in gene expression trait PRS (mean, SD) | Number of trait pairs with shared genetic effect/Number of colocalized trait pairs | Number of trait pairs with significant FDR-adjusted PRS p-value (with shared effect/colocalized) | Mean trait variance explained by gene expression PRS in trait pairs with shared vs. colocalized associations | Difference in trait variance explained by gene expression PRS in shared vs. colocalized associations (Mann–Whitney test p-value) | Evidence of shared association predicting immune trait variance explained by gene expression PRS (univariate regression $R^2$, p-value) |
|---|---|---|---|---|---|---|
| $5 \times 10^{-8}$ | 1.37, 0.69 | 71/6,539 | 10 (14.1%)/75 (1.2%) | $1.00 \times 10^{-02}/1.95 \times 10^{-03}$ | $\mathbf{3.90 \times 10^{-06}}$ | **0.032 [0.024-0.041], $2.11 \times 10^{-49}$** |
| $1 \times 10^{-7}$ | 1.43, 0.75 | 77/7,177 | 10 (13.0%)/78 (1.1%) | $9.46 \times 10^{-03}/1.95 \times 10^{-03}$ | $\mathbf{1.07 \times 10^{-05}}$ | **0.025 [0.018-0.032], $9.62 \times 10^{-42}$** |
| $5 \times 10^{-7}$ | 1.69, 0.97 | 118/10,128 | 8 (6.8%)/62 (0.6%) | $6.06 \times 10^{-03}/1.78 \times 10^{-03}$ | $5.34 \times 10^{-02}$ | **0.014 [0.009-0.018], $3.05 \times 10^{-32}$** |
| $1 \times 10^{-6}$ | 2.00, 1.17 | 132/12,321 | 9 (6.8%)/42 (0.3%) | $4.79 \times 10^{-03}/1.70 \times 10^{-03}$ | $6.55 \times 10^{-02}$ | **0.009 [0.006-0.012], $1.05 \times 10^{-25}$** |
| $5 \times 10^{-6}$ | 4.80, 2.28 | 189/16,293 | 4 (2.1%)/16 (0.1%) | $2.76 \times 10^{-03}/1.53 \times 10^{-03}$ | $7.20 \times 10^{-01}$ | **0.004 [0.002-0.005], $1.01 \times 10^{-14}$** |
| $1 \times 10^{-5}$ | 8.54, 3.07 | 190/16,458 | 2 (1.1%)/4 (0.0%) | $2.13 \times 10^{-03}/1.46 \times 10^{-03}$ | $7.38 \times 10^{-01}$ | **0.002 [0.000-0.003], $7.19 \times 10^{-08}$** |
| $5 \times 10^{-5}$ | 36.47, 6.29 | 190/16,462 | 0 (0.0%)/1 (0.0%) | $2.01 \times 10^{-03}/1.37 \times 10^{-03}$ | $\mathbf{9.76 \times 10^{-03}}$ | **0.001 [0.000-0.002], $1.31 \times 10^{-05}$** |
| $1 \times 10^{-4}$ | 69.08, 8.75 | 190/16,462 | 0 (0.0%)/0 (0.0%) | $1.90 \times 10^{-03}/1.35 \times 10^{-03}$ | $\mathbf{2.59 \times 10^{-03}}$ | **0.001 [0.000-0.001], $7.65 \times 10^{-04}$** |
| $1 \times 10^{-3}$ | 569.59, 26.20 | 190/16,462 | 0 (0.0%)/0 (0.0%) | $1.70 \times 10^{-03}/1.33 \times 10^{-03}$ | $3.04 \times 10^{-01}$ | 0.000 [0.000-0.001], $5.21 \times 10^{-02}$ |
| $1 \times 10^{-2}$ | 4291.77, 72.84 | 190/16,462 | 0 (0.0%)/0 (0.0%) | $1.51 \times 10^{-03}/1.31 \times 10^{-03}$ | $\mathbf{4.29 \times 10^{-02}}$ | 0.000 [0.000-0.000], $1.99 \times 10^{-01}$ |

eQTL/immune trait pairs in our analysis. For each, we identified all independent variants meeting a threshold of association in the eQTL trait. We used these variants to calculate polygenic risk scores (PRS) for each individual in the Milieu Intérieur project immune phenotype collection, and calculated the proportion of paired immune trait variance explained by that PRS ($R^2_{PRS}$). We compared $R^2_{PRS}$ values for trait pairs with and without evidence for a shared underlying genetic effect using a Mann–Whitney–Wilcoxon test and determined the associations of JLIM p-values with $R^2_{PRS}$ values and PRS p-values using univariate linear regression. To account for shared effects between some trait pairs but not others, we conditioned eQTL traits on the variant with the strongest JLIM p-value, and used the now conditionally independent eQTL data for the PRS calculation. Statistically significant results are shown in bold font.

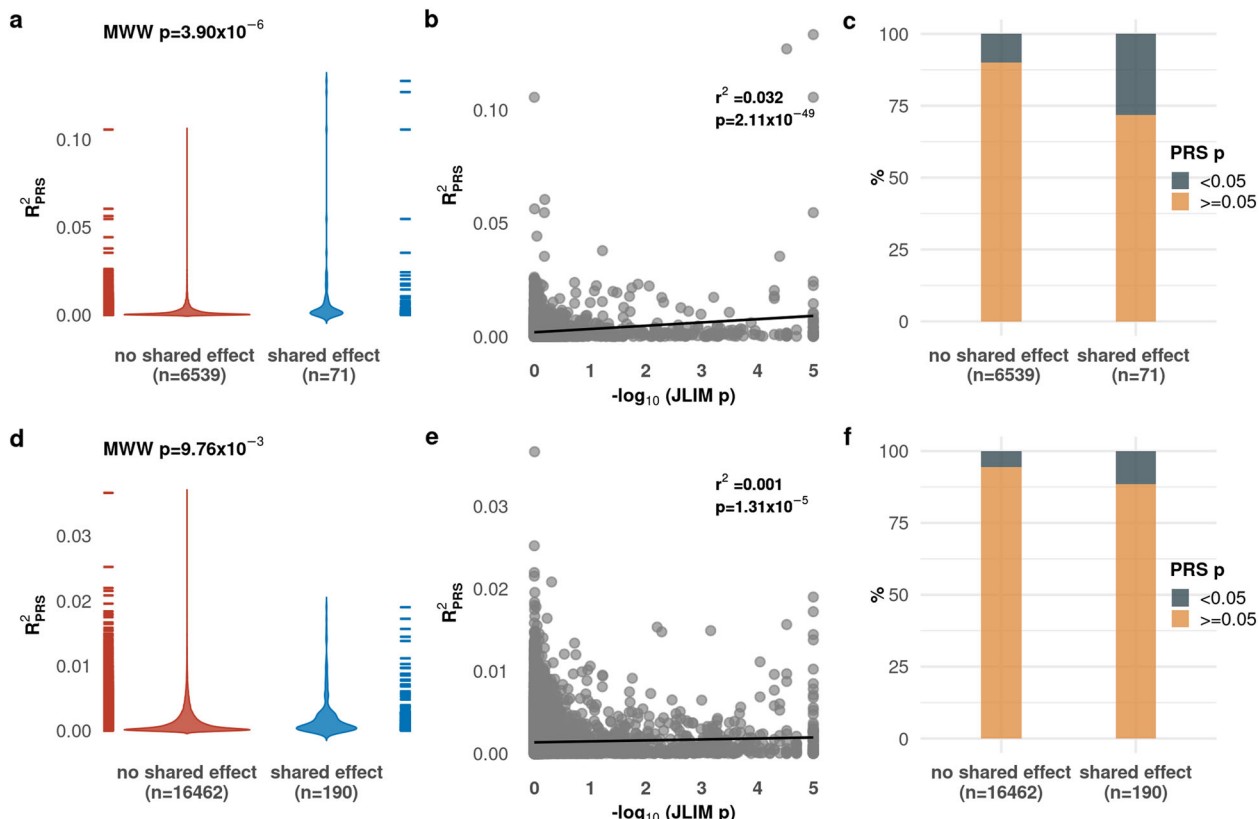

**Fig. 3 The immune phenotype variance explained by gene expression polygenic risk scores (PRS) is higher for trait pairs sharing associations.** We considered 16,652 gene expression/immune trait pairs in our analysis. For each, we identified all independent variants meeting a threshold of association in the gene expression trait. We used these variants to calculate PRS for each individual in the Milieu Intérieur project immune phenotype collection, and calculated the proportion of immune phenotype variance explained by these PRS ($R^2_{PRS}$). We found that gene expression PRS could explain significantly more immune trait variance for gene expression/immune trait pairs with a shared underlying genetic effect than for those that did not share an association. We saw this effect at different thresholds for selecting PRS instruments **a** expression quantitative trait locus (eQTL) $p < 5 \times 10^{-8}$; and **d** eQTL $p < 5 \times 10^{-5}$. We also found that the proportion of variance explained was correlated to the overall strength of evidence for a shared (JLIM $p$-value) at these two selection thresholds (**b**, **e**). The proportion of trait pairs with a $p$-value < 0.05 for the variance of the immune phenotype explained is higher in trait pairs with shared effects (**c**) as compared to trait pairs without a shared genetic effect (**f**). MWW = Mann–Whitney–Wilcoxon test.

suggested[33]. We then perform the same inverse variance-weighted MR analysis as above, using the expanded instrument sets.

We found that gene expression traits explained a higher proportion of immune phenotype variance in the 190 pairs with shared associations, compared to the 15,462 other pairs (with a median absolute estimated causal effect of 0.267 and 0.095, respectively; Fig. 4d). As in our TSMR analysis, we saw a positive correlation between the strength of the JLIM $p$-value and the estimated causal effects of the gene expression traits on immune phenotypes (Fig. 4e). We also saw a significantly higher proportion of trait pairs with a FDR adjusted $p$-value < 0.05 for trait pairs with a shared association (36.8%) as compared to trait pairs without a shared association (12.5%) in this Mendelian randomization analysis (Fig. 4f). Cumulatively, our results suggest that trait pairs with a shared association are more likely to be causally related than trait pairs that do not share an association.

## Discussion
In this work, we report that immune cell traits and gene expression traits share a small but significant set of associations, and that these point to interesting biological events with mechanistic implications. We then show that traits sharing a pleiotropic association tend to be causally related, rather than subject to

horizontal pleiotropy. Thus, even though individual analyses may be under-powered—especially for the likely more complex immune traits—we are able to show in bulk that shared associations can occur between causally related traits. Like all methods, the ones we use here have limitations. All comparisons of genetic mapping results are sensitive to even subtle differences in population structure and consequent differences in LD patterns; and MR approaches, including two-sample methods, can also be sensitive to invalid instruments. We also note that PRS based on small numbers of variants can be unstable, and effects that are clearly not shared can show a substantial proportion of variance explained through LD (Supplementary Figs. 6 and 7).

These shared associations can uncover previously unknown facets of immune cell biology. The shared effect between *SELL* (L-selectin) expression and surface protein levels illustrates this principle: an allele that increases *SELL* transcript levels in neutrophils and monocytes also increases the mean fluorescence intensity of the L-selectin protein product CD62L expressed on neutrophil and eosinophil surface membranes[17]. L-selectin is a cell adhesion molecule used by diverse immune cells to enter target organs by interacting with resident endothelial cells[34]. It is particularly important for the entry of naive T cells into secondary lymphoid tissues as part of the maturation process[35,36]. CD62L levels are thought to be predictive of treatment response in leukemia[37] and risk of adverse events in multiple sclerosis

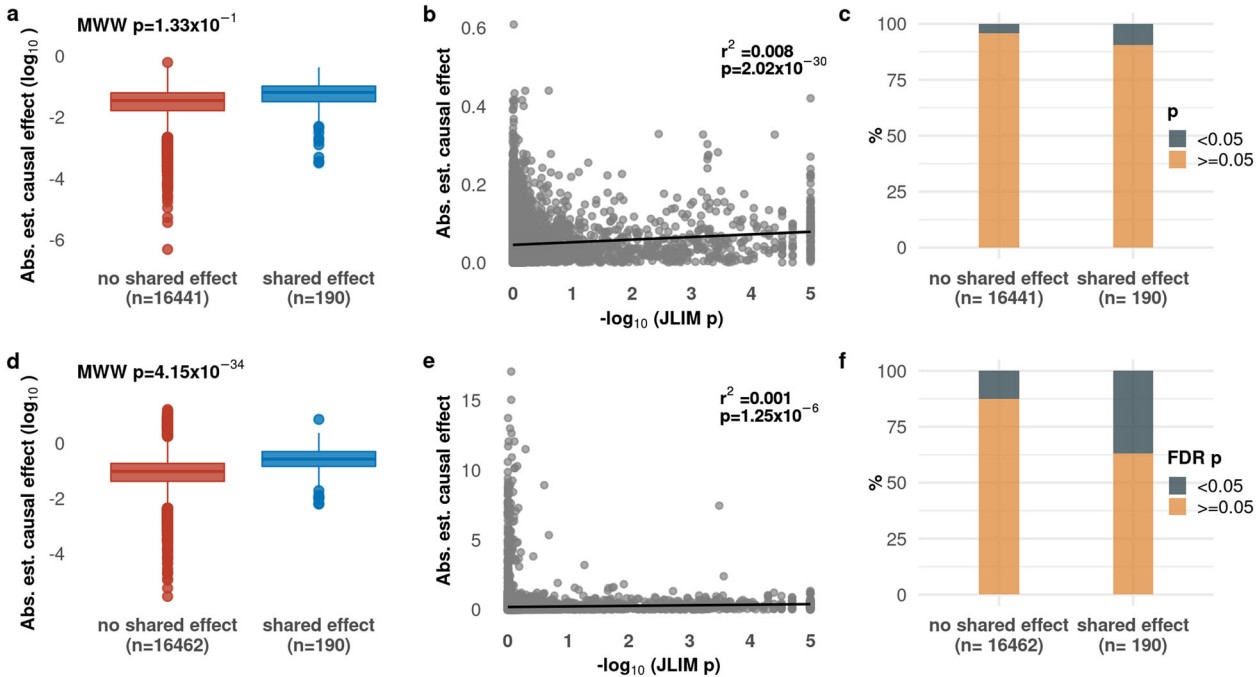

**Fig. 4 Trait pairs sharing an association are more likely to be causally related.** We assessed evidence for causality between gene expression and immune cell traits using Mendelian randomization with cis-eQTL (expression quantitative trait loci) single nucleotide polymorphisms (SNPs) (**a–c**) and an expanded set of instruments including nearby genes (**c–e**). For the latter, we saw a significant level of increased evidence for causal effects in trait pairs with a shared effect (blue) compared to pairs without a shared effect (red) (**d**). We also saw an overall correlation between evidence for shared effects and the estimated causal effect size (**b**, **e**). The percentage of trait pairs with a *p*-value of <0.05 in the TSMR was higher for trait pairs sharing an association (**c**) as was the percentage of trait pairs with an FDR adjusted *p*-value of <0.05 when using an expanded set of instruments (**f**). MWW = Mann–Whitney–Wilcoxon test.

therapy[38]; the effect of the L-selectin eQTL on protein levels could thus be misconstrued in a clinical setting.

We were struck by the number of shared effects between monocyte and neutrophil gene expression traits and T cell immune parameters. In these cases, the genes are either not detected at all, or show no evidence of an eQTL, in our T cell data (Fig. 2a–d, i–l). For example, we found a shared effect between *RPS20P15* in monocytes and the number of naïve CD4 + T cells (Figures d, h, Supplementary Fig. 8). There is no corresponding eQTL in T cells. *RPS20P15* is an expressed ribosomal protein pseudogene encoded in the first exon of *SLC4A7*, and appears to act as an enhancer for that gene[39]. *SLC4A7* encodes a bicarbonate transporter involved in macrophage phagosome acidification[40], and which is activated and expressed during macrophage activation[41]. Bicarbonate induces macrophage response to stimulus[42]. This suggests that differences in *SLC4A7* regulation affect the degree of macrophage activation in response to stimulus, and this has a knock-on effect on T cell homeostasis.

Similarly, we see a shared effect between an eQTL for *CARS2* in neutrophils and CD4/CD8 ratio (Fig. 2c, g, Supplementary Fig. 9). The gene is expressed in CD4 + T cells in our data, and has a separate eQTL mapping approximately 100 kilobases from this shared effect (Fig. 2k). *CARS2* is a nuclear-encoded likely mitochondrial gene, whose product catalyzes cysteine loading onto tRNA. Mutations in this gene have been associated with myoclonic epilepsy, indicating that perturbation has functional consequences[43]. It is tempting to hypothesize that variation in mitochondrial homeostasis in neutrophils leads to changes in T cell homeostasis, either through changes to infection response or baseline cell crosstalk.

Overall, these results suggest that changes to gene expression in one cell type can have a direct effect on another population. Our results also suggest that eQTLs in the three BLUEPRINT cell types also influence traits in other cell types (Supplementary

Figs. 10 and 11); however, as we do not have eQTL data in those cell types, we cannot say with certainty that the same eQTL is not present there, creating a situation analogous to the *SELL* example. Nonetheless, widespread cross-talk between immune cell subsets is a well-attested phenomenon, but to our knowledge this is the first time genetic mapping uncovers mechanisms of cellular coordination across cell subsets. Predictions stemming from these hypotheses will require experimental dissection.

A major challenge in human genetics is translating genotype-phenotype associations into testable hypotheses of the underlying molecular, cellular and physiological events. Directly predicting the effect of a trait-associated variant remains challenging, especially for non-coding polymorphisms. This is further hampered by the limited resolution of fine mapping, so that in most cases we can only narrow an association signal to a group of variants over a genomic interval, all of which must be investigated, rather than pinpoint the exact causal variant. As variant function prediction ability is limited, direct experimentation is necessary, gradually uncovering molecular, cellular and ultimately physiological effects. Without prior information, a variety of outcomes across different cell types and conditions need to be assessed at each stage to uncover trait-relevant events. This approach is not scalable, so most translation efforts are necessarily conducted piecemeal.

One alternative approach to this bottleneck is to exploit pleiotropy across traits to generate molecular, cellular and physiological mechanism hypotheses, which can then be tested experimentally in a more focused way. This approach uses the fact that a variant associated with a physiological trait must act on molecular and cellular events; these are, by definition, also genetic traits as they are altered by a genetic variant, and the variant must have a pleiotropic effect on all these traits. We can thus compare, in unbiased fashion, many molecular traits (gene expression levels, in the present work) with many cellular traits (here, immunological parameters) to identify pleiotropic effects. Two

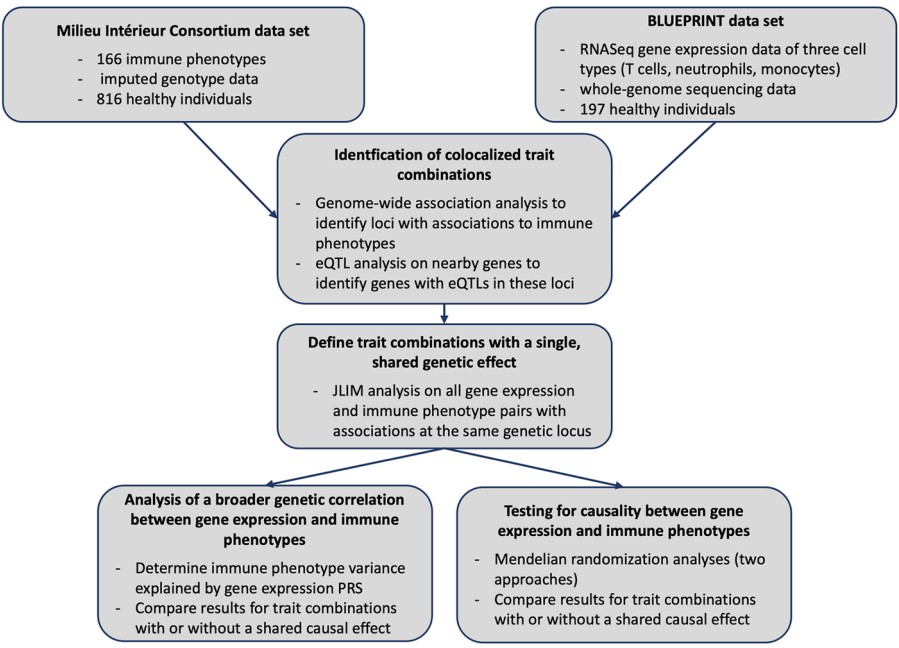

**Fig. 5 Overview of the analysis pipeline steps.** We compared 164 distinct immune cell phenotypes from the Milieu Intérieur project to gene expression traits in monocytes, neutrophils and T cells from the BLUEPRINT consortium. We first selected pairs of immune traits and gene expression traits with associations at the same genetic locus, and then identified which trait pairs share a genetic association and which are associated with different genetic variants in close proximity. Using polygenic risk scores (PRS) we then found that trait pairs with shared genetic associations are more likely to share a broader genetic correlation. Additionally, using two different Mendelian randomization approaches we found that trait pairs that share genetic associations are more likely to share a causal relationship. Abbreviations: eQTL = expression quantitative trait locus, JLIM = Joint Likelihood Mapping.

broad approaches to this cross-trait pleiotropy have been articulated: the first is to look for shared heritability between two traits genome-wide[44,45]; and the second to look for shared effects at specific loci where we see genotype-phenotype associations, as we do in the present work. The latter is particularly suited to traits such as gene expression, where one variant often explains a large proportion of trait variance.

After identifying pairs of traits that share genetic effects (either locus-specific or genome-wide), it is tempting to immediately conclude that one trait directly causes the other. This conclusion is particularly appealing when considering traits across different levels of physiology, as we do here with gene expression and cellular measurements. Our results, however, suggest that simply observing association of different traits at the same genetic locus is not sufficient. Rather, we must look for direct evidence for causality between traits, and shared association mapping is a useful approach to do so. Ultimately, only with explicit proof of causality can we construct mechanistic hypotheses about trait physiology.

## Methods

A schematic overview of our analysis is presented in Fig. 5. Unless otherwise specified, all analyses were carried out with R v3.4.1[46].

**Milieu Intérieur project immune phenotype data processing**. We obtained imputed genotype data and flow cytometry measurements for 166 immune phenotypes (75 innate immune cell parameters, 91 adaptive immune cell parameters; see Supplementary Table 1) for 816 healthy, unrelated people of Western European ancestry from the Milieu Intérieur project[17]. We removed two phenotypes due to the low number of non-zero values (Supplementary Fig. 12a, b). We found that all but one of the remaining phenotypes were not normally distributed (Shapiro-Wilk test), so we performed rank-inverse transformation on all phenotypes. After this transformation, four phenotypes still showed evidence of non-normal distribution. From visual inspection, two of these phenotypes—the number of founder B cells and the number of HLA-DR$^+$/CD4$^+$ EMRA T cells (Supplementary Fig. 12c, d)—appeared not to be detectable in a substantial subset of individuals. We therefore reduced these to binary detected/not detected phenotypes. For each phenotype, we defined the detection threshold as the point of the quantile-quantile plot with the

highest slope (Supplementary Fig. 13). The other two phenotypes had approximately normal distributions despite the Shapiro-Wilk test (Supplementary Fig. 12e, f), so we did not modify them further.

All 816 individuals had been genotyped using the HumanOmniExpress-24 BeadChip and most of them (966 of the initially 1.000 individuals in the cohort) have also been genotyped using the HumanExome-12 Beach Chip and quality control (QC) and genotype imputation has been performed as described in the original publication[17], yielding a final data set of 5,699,237 SNPs with an IMPUTE Score >0.8 and a minor allele frequency (MAF) > 0.05. We performed additional QC, removing all individuals with excess heterozygosity of more than five standard deviations from the sample mean ($n = 6$), one sample from each pair showing cryptic relatedness (identical by descent (IBD) > 0.1875, $n = 3$) and population outliers with a distance in the first four principal components of more than 4 standard deviations ($n = 10$). These QC steps were all based on a set of variants with MAF > 0.05, genotyping rate >98% and a Hardy–Weinberg equilibrium (HWE) test $p$-value > $1 \times 10^{-3}$ and pairwise linkage disequilibrium (LD) < 0.2. From the complete dataset, we then removed all variants out of Hardy–Weinberg equilibrium ($p < 1 \times 10^{-5}$) and MAF < 0.05, and insertions, deletions and multiallelic variants. Our final dataset was thus 5,231,477 variants across 797 individuals.

**BLUEPRINT expression QTL data processing**. We obtained RNA-seq data for naive CD4$^+$ T cells (169 individuals), CD14$^+$ monocytes (193 individuals) and CD16$^+$ neutrophils (196 individuals) from the BLUEPRINT consortium, ascertained to be free of disease and representative of the United Kingdom (UK) population[18]. We downloaded FASTQ files and used the GTEx pipeline for RNA-seq alignment, quantification and quality control (https://www.gtexportal.org/, Analysis Methods for V8). Briefly, we performed alignment to the human reference Genome GRCh38/hg38 using STAR v2.5.3a[47], based on the GENCODE 26 annotation and gene-level quantification with RNA-SeQC v1.1.9[48]. We produced read counts and "transcript per million" (TPM) values as described in the GTEx pipeline. We then selected genes with expression values of >0.1 TPM and ≥6 reads in at least 20% of the samples and normalized between samples using "trimmed mean of M-values" (TMM) as implemented in *edgeR*[49]. We then normalized expression values across samples using an inverse normal transformation. All samples had at least 10 million unique reads. From the BLUEPRINT data release, we obtained genotype data for all individuals for 7,008,524 variants acquired by whole genome sequencing. Sequencing, alignment, variant calling and quality control had been performed as described in the original publication[18]. We additionally filtered out insertion/deletion and multiallelic variants, and all variants with a MAF < 0.05 and a Hardy-Weinberg equilibrium chi-square $p$-value of <1 × $10^{-5}$. We performed sample QC as described above for the MIP dataset, which did

not lead to the removal of any individuals, yielding a final genotype data set of 197 individuals and 4,853,096 single nucleotide polymorphisms (SNPs) (GRCh37 build). In total, we found 4,355,418 SNPs present in both the BLUEPRINT and the Milieu Intérieur project data sets.

Both the BLUEPRINT and the Milieu intérieur project genotype data sets were available in the GRCh37 build, but version 8 of the GTEx pipeline for RNA-seq alignment and quantification uses GRCh38. We reconciled the different genome builds by back-lifting the RNA-seq data to GRCh37, determining the transcription start site for each gene with R/BiomaRt v2.34.3[50,51]. Allele inconsistencies between the two data sets were resolved by transforming the regression coefficients accordingly and ambiguous SNPs were removed for the PRS.

**Association analyses**. We performed all association regression analyses with plink v1.9[52], assuming an additive model of inheritance for all variants. We adjusted all regression analyses on the immune phenotypes from the MIP data set for age, sex, as well as two environmental factors—smoking (0=Non-smoker, 1=Ex-Smoker, 2=Smoker) and latent CMV infection (CMV serology 0=negative, 1=positive)—as these have been identified as the main non-genetic factors affecting immune phenotype variation in the original study[17]. Additionally, we corrected the regression models for the top five principal components to adjust for population stratification. For the association analyses on gene expression data (eQTL analyses) we included age, sex, the first five principal components as well as 30 PEER factors[53] (calculated as described in the GTEx pipeline) as covariates. We used the same covariates to generate permutation data for JLIM.

**Identifying immune and gene expression trait associations in the same locus**. JLIM compares association data for a primary trait to association data for a secondary trait. In all analyses, we use the Milieu Intérieur immune phenotypes as primary traits and BLUEPRINT gene expression traits as secondary. We thus first identify potential associations in the immune phenotypes and then look for overlapping BLUEPRINT eQTLs.

We first identified all independent immune phenotype associations by selecting lead SNPs that (i) had suggestive levels of association ($p < 1 \times 10^{-5}$); (ii) are not within 100 kilobases from another lead SNP; and (iii) are not within 500 kilobases from another lead SNP and in LD ($r^2 > 0.2$) with another lead SNP. To identify conditionally independent associations, we performed stepwise conditional association analyses for all markers within 200 kilobases of each lead SNP. At each step, we selected the most associated SNP not in LD with any other lead SNP ($r^2 < 0.2$); if this SNP had $p < 1 \times 10^{-3}$, we added it to the model and repeated the analysis until no independent SNP satisfied the p-value threshold. For each conditionally independent signal, we then calculated residual association statistics, where we condition on all other independent effects in a locus. All association signals with a lead SNP with an association $p < 1 \times 10^{-5}$ were carried forward to subsequent analyses. These represent strong independent associations, with any residual weak effects removed (identified by the more lenient $p < 1 \times 10^{-3}$ threshold). We deliberately chose lenient thresholds for inclusion to maximize our chances of identifying associations which may be shared across traits.

We next identified cis-eQTLs overlapping immune phenotype associations. We adopted the GTEx definition of a cis-eQTL being within 1 megabase of the transcription start site of the gene. We looked for conditionally independent immune phenotype associations within 200 kilobases of each lead SNP above; we therefore identified all genes with a transcription start site (TSS) within 1 megabase of each lead SNP (R/BiomaRt v2.34.3, Ensembl build 37). We then looked for cis-eQTL associations for each such gene in T cells, monocytes and neutrophils, independently. For each identified eQTL ($p < 1 \times 10^{-3}$), we then performed stepwise conditional association analyses as described above, for all SNPs within 1.2 megabases of the TSS (Supplementary Fig. 14). We chose this distance so any effects overlapping the lead SNP window are conditionally independent.

Due to the smaller sizes of the gene expression traits we limited iterations to a maximum of three independent signals per locus. As above, we then calculated residual association statistics for each independent eQTL effect in each of the three cell types. For each of the genetic loci associated with immune phenotypes we then selected all gene expression association statistics with a lead SNP with an association $p < 1 \times 10^{-3}$ at the respective genetic locus.

**Identifying shared associations between immune and gene expression traits**. We tested for shared effects between immune and gene expression trait pairs with JLIM v2[12]. Given genotype-phenotype associations for two phenotypes in different cohorts in the same locus, JLIM assesses the likelihood of the joint model that variant $i$ is causal in one trait and variant $j$ in another trait, over some number of variants observed in two distinct cohorts. If this joint likelihood is maximal when $i = j$ we can infer the presence of a single, shared effect driving both associations. Conversely, when the likelihood is maximal when $i \neq j$ we can infer that the observed associations are due to different underlying effects. JLIM assumes that only one causal variant for each of the tested traits is present in the analyzed window.

For each trait pair, we used the immune phenotype as the primary trait and the gene expression trait as secondary. We used the 404 non-Finnish European samples from the 1000 Genomes Project (phase 3, release 2013/05/02) as an external LD reference panel. We permuted the secondary trait for each pairwise comparison 100,000 times to obtain empirical significance levels, and used a false discovery rate (FDR) < 0.05 as a significance threshold.

We compared 16,652 unique combinations of immune phenotype and gene expression trait in one of three cell types across 1,199 genetic loci. These pairs encompassed 164 immune traits and 7,060 genes with eQTLs in at least one of the three BLUEPRINT cell types. As we considered up to three conditionally independent associations per gene and locus, we made a total of 22,379 comparisons.

**Calculating polygenic risk scores within trait pairs**. We used polygenic risk scores (PRS) to assess the global genetic overlap between immune/gene expression trait pairs, beyond the shared associations identified by JLIM. For each pair, we selected independent variants associated with the expression trait at some threshold and then calculated PRS for all individuals in the immune trait cohort, using PRSice v2.2.11.b[54] with default parameters for clumping and an additive genetic model. A PRS $\hat{S}_i$ for the $i$th individual over $m$ independent SNPs is defined as:

$$\hat{S}_i = \sum_{i=1}^{m} X_j \hat{\beta}_j$$

where $X_j$ is the number of minor alleles carried at the $j$th SNP, and $\hat{\beta}_j$ is the eQTL effect size for the $j$th SNP[26]. To account for shared effects between some trait pairs but not others, we condition eQTL traits on the main variant from the association signal with the strongest JLIM p-value (even if not significant), and use the now conditionally independent eQTL data for the PRS calculation. We used ten different significance thresholds to select these SNPs: $1 \times 10^{-2}$, $1 \times 10^{-3}$, $1 \times 10^{-4}$, $5 \times 10^{-5}$, $1 \times 10^{-5}$, $5 \times 10^{-6}$, $1 \times 10^{-6}$, $5 \times 10^{-7}$, $1 \times 10^{-7}$, and $5 \times 10^{-8}$. We then calculated the proportion of immune phenotype variance ($R^2$) explained by these PRS and their empirical significance, also using PRSice.

We then compared PRS results between trait pairs with a shared effect and trait pairs with no sharing, using two approaches. We compared the proportion of immune phenotype variance explained (i.e., the $R^2$ values) with the Mann–Whitney–Wilcoxon test, and the correlation between JLIM p-values and PRS $R^2$ with univariate linear regression.

**Mendelian randomization analyses**. We used two Mendelian randomization (MR) approaches to assess evidence that gene expression traits are causal for the immune phenotype traits for which they share an association. First, we used two-sample Mendelian randomization (TSMR)[27,28], as implemented in the *TwoSampleMR v0.4.25*R package, using inverse variance weighting of effect sizes. As instruments, we selected all independent SNPs associated with the gene expression trait with an association p-value < $1 \times 10^{-5}$.

We also used transcriptome-wide summary statistics-based Mendelian Randomization (TWMR), an extension of TSMR[33]. As described by Porcu et al, we first selected all variants associated with the gene expression trait in each trait pair in each cell type (conditional association $p < 1 \times 10^{-3}$). We then identified all genes within 1 megabase of the gene's TSS, and selected all SNPs associated with the expression levels of any of these genes. We then removed genes with highly correlated expression values to the original gene ($r^2 > 0.2$), and selected pairwise-independent SNPs from the remaining list (pairwise LD $r^2 < 0.1$). We used the resulting set of variants as instruments in a multivariate MR model to estimate the causal effect on the immune phenotype. Ambiguous SNPs were removed for both MR analyses.

We compared causality estimates from both methods between trait pairs with a shared effect and trait pairs with no sharing. We compared estimated effect sizes with the Mann–Whitney–Wilcoxon test; and, as a continuous measure, the correlation between JLIM p-values (strength of evidence of shared effect) with the estimated causal effect sizes and corresponding p-values using univariate linear regression.

**Reporting summary**. Further information on research design is available in the Nature Research Reporting Summary linked to this article.

## Data availability
No data were generated beyond the publicly available datasets used. The BLUEPRINT data was retrieved from the European Genome-phenome Archive (EGA) with the following accession numbers: EGAD00001002663[55] (whole genome sequencing), EGAD00001002671[56], EGAD00001002674[57], and EGAD00001002675[58] (RNA-seq data for naive CD4+ T cells, monocytes and neutrophils, respectively). The Milieu Intérieur project genotype data was retrieved from the EGA with the accession number EGAS00001002460[59]. Data underlying Figs. 1–4 are provided in Supplementary Data 1–4, respectively. Supplementary Table 1 is also available as an Excel file in Supplementary Data 5.

## Code availability
Details about software and algorithms used in this study are given in the "Methods" section. All code, including dependency versions, is available on our GitHub repository

(https://github.com/cotsapaslab/immuneMR); the exact version is archived at Zenodo (https://doi.org/10.5281/zenodo.4472530).

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

## Acknowledgements

This study makes use of data generated by the Blueprint Consortium. A full list of the investigators who contributed to the generation of the data is available from www.blueprint-epigenome.eu. Funding for the project was provided by the European Union's Seventh Framework Programme (FP7/2007-2013) under grant agreement no 282510 – BLUEPRINT. We thank the Milieu Intérieur Consortium for kindly providing access to their data. C.G. received a research fellowship from the Deutsche Forschungsgemeinschaft (DFG, German Research Foundation) for this project. She further received funding from the Hans und Klementia Langmatz-Stiftung and the Hertie Network of Excellence in Clinical Neuroscience, not related to this study.

## Author contributions

All authors designed research. S.C. and S.R.S. contributed software. C.G. and C.C. analyzed data and wrote the manuscript, which was approved by S.C. and S.R.S.

## Competing interests

The authors declare no competing interests.
