## [Peer Review File · Communications Biology]

Reviewers' comments:

Reviewer #1 (Remarks to the Author):

In the manuscript COMMSBIO-20-2729 entitled "Shared associations identify causal relationships between gene expression and immune cell phenotypes", Gasperi and colleagues test whether shared genetic associations between immune phenotypes and gene expression levels are observed because the latter traits cause the former (SNP \rightarrow expression \rightarrow immune phenotype), or because of horizontal pleiotropy (expression \leftarrow SNP \rightarrow immune phenotype). Their more general objective is to use statistical modelling to generate mechanistic hypotheses about trait physiology. To do so, they analyzed publically available data from the Milieu Intérieur and BLUEPRINT Consortia, and used statistical methods to find shared causal variants, and test causality, between immune phenotypes and gene expression, using polygenic scores and two-sample Mendelian randomization. They find that a minority of trait pairs share a causal variant (0.9%). They highlight a number of relevant examples where causality is clear, such as *SELL* gene expression and CD62L protein levels in granulocytes, and other, more complex examples. They show that polygenic scores of gene expression explain more of the variance of immune phenotypes when they share causal variants. Furthermore, they show that causal effects of gene expression traits on immune phenotypes are higher for pairs with a shared association.

I find that the objectives, methods and results of the study are clear and straightforward. The study is of interest, because it provides evidence that causal inference has an important role to play in translating genome-wide significant association signals into meaningful biological mechanisms. I have, however, major comments about the interpretation of the results.

Major comments:

- The authors report strong statistical evidence that some trait pairs with a shared association also show higher genetic correlation or TSMR causal effects. These are probably true positive signals, which are informative about physiological mechanisms underlying genotype-phenotype associations. However, the very large majority of trait pairs with a shared association do not show such patterns (Figs. 3 and 4). I suspect that correlations between R^2_{PRS} and $-\log_{10}(P_{\text{JLIM}})$, or between TSMR causal effects and $-\log_{10}(P_{\text{JLIM}})$, as well as MWW significance, are driven by a relatively low number of strong, true causal effects. Consequently, I suggest that the authors (1) comment further such true signals, and (2) report, among JLIM significant pairs, the proportion of pairs with significant R^2_{PRS} or TSMR causal effects. I detail my suggestions below.

(1) The authors highlight some strong case examples of trait pairs with shared associations where causality is clear. One example is *SELL* gene expression and CD62L protein levels in granulocytes, which is very relevant, but already reported in a previous study based on the same data (Patin et al., *Nat Immunol* 2018). I note that they also report intriguing new observations about the direction of the *SELL* eQTL in T cells. Nevertheless, I invite the authors to explore further the strongest signals, and to highlight other new promising examples. In addition, the authors rightfully highlight more complex examples where the pair includes an immune cell trait and a gene expression trait in another cell type, which suggests an interplay between cell subpopulations. Again, I invite the authors to explore further these examples, which may generate interesting hypotheses in immunology. In other words, the general goal of the study is to make use of statistical tools to generate such mechanistic hypotheses; therefore, the authors should comment in more detail their results.

(2) The authors conclude that evidence of shared association is not evidence for causality, which is

not, if I may say, a very novel conclusion, and is not clearly demonstrated in the study. Nevertheless, they have everything they need to estimate the proportion of trait pairs with (or without) shared associations (JLIM significant pairs) that also show a significant R^2 PRS or TSMR causal effect. These proportions would be a very valuable information, which should be reported and commented (as the authors did for the proportion of trait pairs sharing a causal variant). Depending on the results, the authors may revise their statements that "most [trait pairs are] due to causal relationships rather than pleiotropy", in the Abstract and the Main Text (P.4 L.20; P.16 L.6-7).

- The authors suggest an interplay between cell types, based on statistical evidence that genetic variants affect both an immune cell trait and gene expression in another cell type. I invite the authors to explore further this hypothesis, by using cell types as primary and secondary traits in MR models (SNP -> cell type 1 -> cell type 2).

- Inspection of immune cell traits measured by Milieu Intérieur suggests that several of these traits are correlated. Therefore, the authors compare distributions of statistics for trait pairs that are not independent. I understand that this is difficult to account for. One possibility is to check that immune traits with shared association with gene expression are not more correlated among them than immune traits without shared associations.

Minor comments:

- The Discussion should include a description of the limitations of the methods used, particularly TSMR.
- P.3, L. 21-27: Another explanation that should be mentioned is the fact that there are numerous eQTLs in the genome, the most of which are not related to the primary traits.
- P.5 L.20: When reading Patin et al., *Nat Immunol* 2018 in detail, it seems that SNP genotyping was done on both the Illumina HumanOmniExpress-24 and HumanExome-12 arrays.
- P.6 L.9: Please replace "CRCh38/hg38" by "GRCh38/hg38".
- P.6. L.23: Please clarify how allele inconsistencies between datasets were resolved.
- P.7 L.5: Please replace "infections" by "infection".
- P.12 L.1: 207 or 190?
- P.12 L.15: Replace "effect" by "allele"?
- P.16 L.10-13: Please cite Patin et al., *Nat Immunol* 2018.
- P.18: Please ask the Milieu Intérieur consortium if they should be acknowledged.

Reviewer #2 (Remarks to the Author):

This is a nice idea to explore the architecture of protein/cell level traits together with eQTLs. It's an area that I think merits a lot of exploration as it's a gap in understanding disease or other complex trait associations at a cellular level.

I have no major comments, though I do worry a bit that the sample size of the MI dataset may be too small to be very well powered (see also my comments about thresholds below). Nothing to do about that here, but perhaps the authors could comment on it.

Minor comments:

1. Code should be available. The statement, "Details about software and algorithms used in this study are given in the "Methods" section. No customized code or algorithm deemed central to the conclusion

was used." is, frankly, ridiculous. Post the scripts used to do the analysis somewhere.

2. Why $p < 1e-5$ in the MI and $< 1e-3$ in the BLUEPRINT? These seem awfully lenient, especially in fairly small sample sizes, where one might be even less convinced that p values in that range are true effects. The authors do show a few different thresholds in Fig 3, but overall it seems unjustified. e.g. the first result in Fig 1 has a g-w sig MI effect and eQTLs with pvalues like $1e-60$. And that's no surprise: stronger results are easier to interpret.

3. Figure 2 is nice, and a point that I think isn't fully appreciated.

4. Could the authors confirm what they mean in the PRS analysis when they say, "We accounted for this bias by performing conditionally independent association testing for the main cis-eQTL effect."?

5. How much variance of the gene expression do the PRS in Figure 3 explain? Is there any correlation between that and the variance explained in the immune traits?

6. There are a few outliers in Figure 3a in terms of variance explained. Is there anything interesting going on there?

We thank the reviewers for their attention to our work and constructive feedback. Here, we respond to each comment inline, quoting changes to the main text as appropriate. We have also made all changes to the main text in the same blue font to make them easier to track.

General comments:

As part of this revision, we noticed a small error in our PRS calculations. PRSice-2 treats binary and continuous phenotypes differently (logistic vs linear regression). Two of our 164 immune phenotypes are binary, but we had not used the logistic framework for these. We now report the correct statistics in Tables 1 and S2. There is no meaningful difference in statistics or interpretation: for example, the p-value for the MWW test for difference in variance explained by JLIM+ vs JLIM- effects selected at a genome-wide significant threshold changes from 4.06×10^{-6} to 3.90×10^{-6} .

Reviewers' comments:**Reviewer #1 (Remarks to the Author):**

In the manuscript COMMSBIO-20-2729 entitled "Shared associations identify causal relationships between gene expression and immune cell phenotypes", Gasperi and colleagues test whether shared genetic associations between immune phenotypes and gene expression levels are observed because the latter traits cause the former (SNP -> expression -> immune phenotype), or because of horizontal pleiotropy (expression <- SNP -> immune phenotype). Their more general objective is to use statistical modelling to generate mechanistic hypotheses about trait physiology. To do so, they analyzed publically available data from the Milieu Intérieur and BLUEPRINT Consortiums, and used statistical methods to find shared causal variants, and test causality, between immune phenotypes and gene expression, using polygenic scores and two-sample Mendelian randomization. They find that a minority of trait pairs share a causal variant (0.9%). They highlight a number of relevant examples where causality is clear, such as SELL gene expression and CD62L protein levels in granulocytes, and other, more complex examples. They show that polygenic scores of gene expression explain more of the variance of immune phenotypes when they share causal variants. Furthermore, they show that causal effects of gene expression traits on immune phenotypes are higher for pairs with a shared association.

I find that the objectives, methods and results of the study are clear and straightforward. The study is of interest, because it provides evidence that causal inference has an important role to play in translating genome-wide significant association signals into meaningful biological mechanisms. I have, however, major comments about the interpretation of the results.

We thank the reviewer for their kind comments and thorough review of our manuscript.

Major comments:

- The authors report strong statistical evidence that some trait pairs with a shared association also show higher genetic correlation or TSMR causal effects. These are probably true positive signals, which are informative about physiological mechanisms underlying genotype-phenotype associations. However, the very large majority of trait pairs with a shared association do not show such patterns (Figs. 3 and 4). I suspect that correlations between R^2_{PRS} and $-\log_{10}(P_{JLIM})$, or between TSMR causal effects and $-\log_{10}(P_{JLIM})$, as well as MWW significance,

are driven by a relatively low number of strong, true causal effects. Consequently, I suggest that the authors

- (1) comment further such true signals, and
- (2) report, among JLIM significant pairs, the proportion of pairs with significant R^2_{PRS} or TSMR causal effects. I detail my suggestions below.

We thank the reviewer for their careful assessment of our work. We detail our response to the first point about commenting on true signals below, in response to their invitation to explore our results further. For the second point, we now report the proportion of trait pairs with significant results in the PRS analysis and the MR analyses in Figures 3 and 4, in tables 1 and S2 as well as in the main text (P.14, L.2-4, P.14 L.32-34, and P.15 L.19-21). We observe significant overall genetic correlation as determined by means of the PRS score analyses for higher proportion of JLIM significant pairs as compared to pairs without a shared genetic effect. The proportion of JLIM significant trait pairs with significant R^2_{PRS} depends on the p-value threshold used for the selection of SNPs to be included in the PRS analysis and is highest for a genome-wide p-value threshold (5×10^{-8}) where it is 14.1%. In the TSMR we did not observe any trait pairs with a significant TSMR p-value after FDR-correction. However, that proportion of trait combinations with a nominally significant TSMR p-value (<0.05) was higher for trait pairs with a shared genetic effect (9.5%) as compared to trait combinations without a shared effect (4.3%). In the second MR analysis we performed (using an expanded set of instruments) we observed that 36.8% of the JLIM significant trait pairs showed significant evidence for causality (FDR adj. p-value < 0.05). However, as we state in the Discussion of our manuscript, the individual analyses may be underpowered to provide sufficient evidence for causality between traits and we therefore show in bulk that shared associations occur between causally related traits.

- The authors highlight some strong case examples of trait pairs with shared associations where causality is clear. One example is SELL gene expression and CD62L protein levels in granulocytes, which is very relevant, but already reported in a previous study based on the same data (Patin *et al.*, *Nat Immunol* 2018). I note that they also report intriguing new observations about the direction of the SELL eQTL in T cells. Nevertheless, I invite the authors to explore further the strongest signals, and to highlight other new promising examples.

We thank the reviewer for their positive comments on our results. They will appreciate that there is a delicate balance between exploration and speculation, and we are wary of falling into the latter. We have expanded the discussion to include several of the hits seen in Figure 2 (P.16 L.29 – P. 17 L.11, supplementary figures S12 and S13). We now also show two examples where eQTLs in the three BLUEPRINT cell types also influence traits in other cell types (P.17 L.13-17, supplementary figures S14 and S15).

“For example, we found a shared effect between RPS20P15 in monocytes and the number of naïve CD4+ T cells (Figures 2, S12). There is no corresponding eQTL in T cells. *RPS20P15* is an expressed ribosomal protein pseudogene encoded in the first exon of *SLC4A7*, and appears to act as an enhancer for that gene⁴⁸. *SLC4A7* encodes a bicarbonate transporter involved in macrophage phagosome acidification⁴⁹, and which is activated and expressed during macrophage activation⁵⁰. Bicarbonate induces macrophage response to stimulus⁵¹. This suggests that differences in *SLC4A7* regulation affect the degree of macrophage activation in response to stimulus, and this has a knock-on effect on T cell homeostasis.

Similarly, we see a shared effect between an eQTL for *CARS2* in neutrophils and CD4/CD8 ratio (Figures 2, S13). The gene is expressed in CD4+ T cells in our data, and has a separate eQTL mapping approximately 100 kilobases from this shared effect (Figure 2). *CARS2* is a nuclear-encoded likely mitochondrial gene, whose product catalyzes cysteine loading onto tRNA. Mutations in this gene have been associated with myoclonic epilepsy, indicating that perturbation has functional consequences⁵². It is tempting to hypothesize that variation in mitochondrial homeostasis in neutrophils leads to changes in T cell homeostasis, either through changes to infection response or baseline cell crosstalk.

Overall, these results suggest that changes to gene expression in one cell type can have a direct effect on another population. Our results also suggest that eQTLs in the three BLUEPRINT cell types also influence traits in other cell types (Figures S14 and S15); however, as we do not have eQTL data in those cell types, we cannot say with certainty that the same eQTL is not present there, creating a situation analogous to the *SELL* example.”

In addition, the authors rightfully highlight more complex examples where the pair includes an immune cell trait and a gene expression trait in another cell type, which suggests an interplay between cell subpopulations. Again, I invite the authors to explore further these examples, which may generate interesting hypotheses in immunology. In other words, the general goal of the study is to make use of statistical tools to generate such mechanistic hypotheses; therefore, the authors should comment in more detail their results.

Please see answer immediately above.

- The authors conclude that evidence of shared association is not evidence for causality, which is not, if I may say, a very novel conclusion, and is not clearly demonstrated in the study. Nevertheless, they have everything they need to estimate the proportion of trait pairs with (or without) shared associations (JLIM significant pairs) that also show a significant R^2 PRS or TSMR causal effect. These proportions would be a very valuable information, which should be reported and commented (as the authors did for the proportion of trait pairs sharing a causal variant). Depending on the results, the authors may revise their statements that "most [trait pairs are] due to causal relationships rather than pleiotropy", in the Abstract and the Main Text (P.4 L.20; P.16 L.6-7).

We now report these results in the manuscript, as detailed above. We have amended the Abstract and Discussion sentences to which the reviewer refers as follows:

“We find that only a subset of traits share associations, with *many* due to causal relationships rather than pleiotropy.” (P.2 L.9)

“we are able to show in bulk that shared associations *can* occur between causally related traits.” (P.16 L.6-7)

- The authors suggest an interplay between cell types, based on statistical evidence that genetic variants affect both an immune cell trait and gene expression in another cell type. I invite the authors to explore further this hypothesis, by using cell types are primary and secondary traits in MR models (SNP -> cell type 1 -> cell type 2).

Our analyses suggest an interplay between cell types as we observe shared genetic association between gene expression traits in one cell type and immune phenotypes measured in another cell type. We further investigate this by means of MR analyses where we observe evidence for

causality between some of these trait pairs in different immune phenotypes. We do not understand exactly which kind of further analysis the reviewer suggests and therefore ask for clarification. However, as mentioned in the manuscript and above we believe that causality analyses for individuals trait pairs might be under-powered in this study.

- Inspection of immune cell traits measured by Milieu Intérieur suggests that several of these traits are correlated. Therefore, the authors compare distributions of statistics for trait pairs that are not independent. I understand that this is difficult to account for. One possibility is to check that immune traits with shared association with gene expression are not more correlated among them than immune traits without shared associations.

We thank the reviewer for this suggestion. We have performed the correlation analysis, and see that immune traits with shared associations with gene expression traits are not more correlated among them than immune traits without such shared association (Spearman's $\rho = 0.11 \pm 0.13$ and 0.13 ± 0.14 , respectively). We have added this information to the results section on page 12 lines 8-12 as:

“We see correlation between immune traits across individuals, suggesting that some gene expression traits may spuriously share associations with more than one such trait; however, we found no difference in correlation between immune traits that shared associations with gene expression traits and immune traits that did not (Spearman's $\rho = 0.11 \pm 0.13$ and 0.13 ± 0.14 , respectively), suggesting this is not a major factor in our analysis.”

Minor comments:

- The Discussion should include a description of the limitations of the methods used, particularly TSMR.

We agree, and have added the following to page 16 line 8-13:

“Like all methods, the ones we use here have limitations. All comparisons of genetic mapping results are sensitive to even subtle differences in population structure and consequent differences in LD patterns; and MR approaches, including two-sample methods, can also be sensitive to invalid instruments. We also note that PRS based on small numbers of variants can be unstable, and effects that are clearly not shared can show a substantial proportion of variance explained through LD (Figures S10 and S11).”

- P.3, L. 21-27: Another explanation that should be mentioned is the fact that there are numerous eQTLs in the genome, the most of which are not related to the primary traits.

In the previous work we cited, each trait association is tested against each eQTL independently. The number of unrelated eQTLs in the genome is therefore not a primary driver of the observed lack of sharing.

- P.5 L.20: When reading Patin *et al.*, Nat Immunol 2018 in detail, it seems that SNP genotyping was done on both the Illumina HumanOmniExpress-24 and HumanExome-12 arrays.

We now report in the Methods section (P.5 L.20-22) that 966 of the 1.000 individuals initially included in the MIP cohort were also genotyped using the HumanExome-12 array. In our study, we did not include rare variants (only SNPs with a MAF > 0.05); however, the HumanExome-12 data was used for the genotype imputation as described in Patin *et al.*

- P.6 L.9: Please replace "CRCh38/hg38" by "GRCh38/hg38".

We thank the reviewer for pointing this out and changed it accordingly (P.6 L.7).

- P.6. L.23: Please clarify how allele inconsistencies between datasets were resolved.

Where necessary, allele inconsistencies between data sets were resolved by transforming the effect sizes (coefficients) accordingly. We now state this in the Methods section (P.6, L.29-30). We also remove ambiguous SNPs from the TWMR analysis which we state in The Methods section (P.10 L.6) and updated the results accordingly (P.15 L.16, Figure 4).

- P.7 L.5: Please replace "infections" by "infection".

We thank the reviewer for pointing this out and changed it accordingly (P.7 L.2).

- P.12 L.1: 207 or 190?

The correct number is 207 here. The confusion arises because for some for some the gene expression / immune trait combinations we could identify more than one locus with a shared association. In total we identified 207 share association signals across 192 unique gene expression / immune phenotype combinations. We have clarified this by adding the following test (P.11 L.32 – P.12 L.3).

“These 207 shared effects include 15 instances where a gene expression and immune phenotype pair shared two (13 pairs) or three (two pairs) genetic effects. For 190 distinct combinations of a gene expression trait and an immune phenotype at least one shared association could be identified. For 190 distinct combinations of a gene expression trait and an immune phenotype at least one shared association could be identified.”

- P.12 L.15: Replace "effect" by "allele"?

We thank the reviewer for pointing this out and changed it accordingly. (P.12 L.20)

- P.16 L.10-13: Please cite Patin *et al.*, Nat Immunol 2018.

We now cite Patin *et al.* (P.16. L.19)

- P.18: Please ask the Milieu Intérieur consortium if they should be acknowledged.

We thank the reviewer for pointing this out. After consultation with the Milieu Interieur team, we have included the following to our acknowledgements (P.19 L.6): “We thank the Milieu Intérieur Consortium for kindly providing access to their data.”

Reviewer #2 (Remarks to the Author):

This is a nice idea to explore the architecture of protein/cell level traits together with eQTLs. It's an area that I think merits a lot of exploration as it's a gap in understanding disease or other complex trait associations at a cellular level.

I have no major comments, though I do worry a bit that the sample size of the MI dataset may be too small to be very well powered (see also my comments about thresholds below). Nothing to do about that here, but perhaps the authors could comment on it.

We thank the reviewer for this comment and agree with this statement. We state in the Discussion section, that individual analyses are under-powered in this study, but that we are able to show in bulk that a substantial proportion of shared associations between traits is likely to be

due to causal relationships. We have added the following text to directly address the MI power issue (P.16 L.6): *“especially for the likely more complex immune traits.*

Minor comments:

1. Code should be available. The statement, "Details about software and algorithms used in this study are given in the “Methods” section. No customized code or algorithm deemed central to the conclusion was used." is, frankly, ridiculous. Post the scripts used to do the analysis somewhere.

The reviewer is correct and we apologize for the omission. We now provide all the scripts used for this study on GitHub (<https://github.com/cotsapaslab/immuneMR>) and added this information to the Code Availability section (P.10 L.22-23).

2. Why $p < 1e-5$ in the MI and $< 1e-3$ in the BLUEPRINT? These seem awfully lenient, especially in fairly small sample sizes, where one might be even less convinced that p values in that range are true effects. The authors do show a few different thresholds in Fig 3, but overall it seems unjustified. e.g. the first result in Fig 1 has a g-w sig MI effect and eQTLs with p-values like $1e60$. And that's no surprise: stronger results are easier to interpret.

We agree that strong associations in both traits will maximize power to detect shared associations. We deliberately set lenient thresholds to ensure we were casting a wide net of possibly shared effects (and as the reviewer points out, also look at the effect in Figure S6). As described above, we include method limitations in the Discussion section. We now justify these thresholds in the Methods section (P.7 L.29-30) as:

“We deliberately chose lenient thresholds for inclusion to maximize our chances of identifying associations which may be shared across traits.”

3. Figure 2 is nice, and a point that I think isn't fully appreciated.

We thank the reviewer for this comment. As stated in our replies to Reviewer 1, above, we now expand on some of these results in the Discussion.

4. Could the authors confirm what they mean in the PRS analysis when they say, "We accounted for this bias by performing conditionally independent association testing for the main cis-eQTL effect."?

We apologize for the lack of clarity and have now amended this statement to read: *“We accounted for this bias by conditioning the eQTL data on the main cis-eQTL effect with the strongest evidence for being shared with the immune phenotype, thus removing the effect of the shared variant (see Methods for details).”* (P.13 L.28-30)

5. How much variance of the gene expression do the PRS in Figure 3 explain? Is there any correlation between that and the variance explained in the immune traits?

We thank the reviewer for raising this interesting question. For 190 expression/immune trait pairs with evidence of a shared effect, we calculated the proportion of variance of gene expression explained by the PRS instruments at different p-value thresholds. The mean proportions of gene expression variance captured are between 0.14 and 0.35 across a range of p-value thresholds

for instrument selection. As the reviewer suspects, we see correlation between gene expression variance captured and immune trait variance captured (Table R1). However, we feel there is some circularity to these calculations as we select instruments and calculate the proportion of variance explained in the same data, which can lead to positive bias. Also, the power to detect this correlation depends on the total amount of immune trait variance explained by each gene expression trait. Thoroughly investigating these phenomena would be a substantial undertaking in itself and somewhat beyond the scope of the present manuscript, so we prefer not to include these potentially biased calculations in the paper.

p-value threshold for PRS calculation	Number of gene expression / immune trait pairs with a shared effect for which a gene expression PRS could be calculated	R^2_{PRS} for gene expression	R^2_{PRS} for the immune phenotypes	Spearman correlation between R^2_{PRS} for gene expression and R^2_{PRS} for immune phenotype	
				p-value	ρ
5×10^{-8}	71	0.18 ± 0.14	$1.00 \times 10^{-2} \pm 2.54 \times 10^{-2}$	0.161	0.168
1×10^{-7}	77	0.18 ± 0.14	$9.46 \times 10^{-2} \pm 2.45 \times 10^{-2}$	7.52×10^{-2}	0.204
5×10^{-7}	118	0.14 ± 0.12	$6.06 \times 10^{-3} \pm 1.81 \times 10^{-2}$	1.52×10^{-3}	0.289
1×10^{-6}	132	0.14 ± 0.12	$4.79 \times 10^{-3} \pm 1.37 \times 10^{-2}$	1.14×10^{-4}	0.330
5×10^{-6}	189	0.16 ± 0.11	$2.76 \times 10^{-3} \pm 6.83 \times 10^{-3}$	2.60×10^{-2}	0.162
1×10^{-5}	190	0.21 ± 0.12	$2.13 \times 10^{-3} \pm 4.50 \times 10^{-3}$	7.78×10^{-2}	0.128
5×10^{-4}	190	0.29 ± 0.14	$2.01 \times 10^{-3} \pm 3.19 \times 10^{-3}$	1.46×10^{-1}	0.106
0.0001	190	0.31 ± 0.15	$1.90 \times 10^{-3} \pm 3.16 \times 10^{-3}$	9.06×10^{-1}	8.59×10^{-3}
0.001	190	0.34 ± 0.16	$1.70 \times 10^{-3} \pm 2.46 \times 10^{-3}$	1.82×10^{-3}	-0.225

0.01	190	0.35 ± 0.17	$1.51 \times 10^{-3} \pm 1.83 \times 10^{-3}$	$6,14 \times 10^{-1}$	0.0369
------	-----	-----------------	---	-----------------------	--------

Table R1: Variance of the gene expression traits and the immune phenotypes explained by the (gene expression) PRS and correlation between the variance of gene expression traits and the variance of the immune phenotypes explained by the PRS.

6. There are a few outliers in Figure 3a in terms of variance explained. Is there anything interesting going on there?

We have previously looked at these outliers (two examples are now shown in new supplementary figures S10 and S11) but had not included this discussion for the sake of space. We generally find that there is clear absence of shared effects in those outliers. We suspect the R^2_{PRS} outlier values may be due to winner's curse, though this is hard to prove in the present analysis. We now include this discussion on the limitations of PRS approaches in the Discussion (P.16 L.11-13) as:

"We also note that PRS based on small numbers of variants can be unstable, and effects that are clearly not shared can show a substantial proportion of variance explained through LD (Figures S10 and S11)."

REVIEWERS' COMMENTS:

Reviewer #1 (Remarks to the Author):

The authors have satisfactorily addressed all my comments. I note that no scripts are yet accessible on their GitHub repository.

Reviewer #2 (Remarks to the Author):

The authors have answered my concerns.